# Upper ocean changes with hurricane-strength wind events: a study using Argo profiles and an ocean reanalysis

Jacopo Sala[1], Donata Giglio[1], Addison Hu[2], Mikael Kuusela[2], Kimberly M. Wood[3], and Ann B. Lee[2]

[1]University of Colorado Boulder, Department of Atmospheric and Oceanic Sciences, Boulder, CO, USA
[2]Carnegie Mellon University, Department of Statistics and Data Science, Pittsburgh, PA, USA
[3]University of Arizona, Department of Hydrology and Atmospheric Sciences, Tucson, AZ, USA

**Correspondence:** Jacopo Sala (jacopo.sala@colorado.edu)

**Abstract.** As the Earth's climate is warming, the intensity and rain rate of tropical cyclones (TCs) is projected to increase. TCs intensify by extracting heat energy from the ocean, hence a better understanding of upper ocean changes with the TC passage is helpful to improve our understanding of air-sea interactions during and after the event. This work uses Argo float observations and the HYCOM ocean reanalysis to describe characteristics of upper ocean changes with hurricane-strength wind events. We study the association of these changes with the vertical structure of the salinity profile before the event, i.e.,
increasing versus decreasing with depth. We also study the contribution of changes in salinity to upper ocean density changes in each case. Results show that in regions where pre-event salinity increases with depth there is a corresponding statistically significant increase in upper ocean salinity; vice versa, we observe a significant decrease in upper ocean salinity in regions where pre-event salinity decreases with depth. Consistent with previous studies, temperature decreases in both regions. As
near-surface temperature decreases, upper ocean density increases and the increase is larger where pre-event salinity increases with depth. Changes in upper ocean properties from Argo and HYCOM are overall consistent with wind-driven vertical mixing of near-surface waters with colder and higher (or lower) salinity waters below. Resulting changes in ocean stratification have implications for air-sea interactions during and after the event, with potential impacts on weather events that follow.

## 1 Introduction

Tropical cyclones (TCs) represent a severe threat for both the human population and the fragile ecosystems that live in the tropical ocean (Greening et al., 2006; Orr and Ogden, 1992), causing thousands of fatalities and billions of US dollars in damage globally each year (Emanuel, 2003; Klotzbach et al., 2018; Mendelsohn et al., 2012).

The number of hurricanes decreased during 1990-2021, likely due to a more La Niña-like base state favoring TC activity in the North Atlantic and suppressing it in the Pacific (Klotzbach et al., 2022); yet TC intensification rates and the frequency
of rapid TC-intensification have increased in the last few years (Balaguru et al., 2018; Bhatia et al., 2019; Kishtawal et al., 2012). As TCs are becoming more powerful (Shaw et al., 2022), it is important to improve our understanding of the physical mechanisms that may regulate air-sea interactions during and after the TC passage (Cione et al., 2013), e.g., to improve the prediction of TC intensification and mitigate the TC-related destructive potential. TCs intensify by extracting energy from the ocean in the form of latent heat (e.g., Chan, 2005; Emanuel, 1999), leading to a substantial ocean cooling (Shen and Ginis,

2003; Han, 2023). Hence the underlying sea surface temperatures (SSTs) are a key factor in determining how TCs evolve in time. Also, the strong winds of tropical cyclones induce upper ocean mixing of cold water from deeper in the ocean with the warm surface waters, a process called upwelling (Webb and Suginohara, 2001; Prasad and Hogan, 2007; Han et al., 2024). This mixing entrains colder water from below the mixed layer and decreases SST (Cui et al., 2023; Elsberry et al., 1976; Fisher, 1958; Karnauskas et al., 2021; Leipper, 1967; Price, 1981), resulting in a negative feedback on the cyclone's intensity via the impact of SST cooling on air–sea enthalpy fluxes (e.g., Balaguru et al., 2012; Bender and Ginis, 2000; Cione and Uhlhorn, 2003; Karnauskas et al., 2021; Lin et al., 2005; Liu et al., 2007; Lloyd and Vecchi, 2011; Prasad and Hogan, 2007; Schade and Emanuel, 1999; Shay et al., 2000). Wind driven vertical mixing is an important mechanism regulating upper ocean changes for all high wind events, including those that are not associated with tropical cyclones (Cardona and Bracco, 2012; Kuang et al., 2011; Large et al., 1994; Meng et al., 2020).

While several studies have investigated cyclone-induced changes in upper ocean temperature and the role the ocean thermal structure plays in the intensification of cyclones (e.g., Zhang et al., 2021), cyclone-induced changes in upper ocean salinity and the role of the vertical structure of the salinity profile in regulating TC-induced changes in upper ocean stratification are not as well understood. This limits our understanding of air-sea interactions during and after the TC passage, as the degree of stratification of the water column is related to vertical gradients of both temperature and salinity and it regulates air-sea exchanges of heat (e.g., Reul et al., 2014b; Vincent et al., 2014) and vertical mixing induced by atmospheric disturbances, including hurricanes (Holyer et al., 1987; Neetu et al., 2012). As an example, Balaguru et al. (2020) show how taking into account both temperature and salinity improves the prediction of TCs' rapid intensification. Satellite-based observations of sea surface salinity show salinification (a salty wake) after TC-passage (Chaudhuri et al., 2019; Grodsky et al., 2012; Liu et al., 2020; Reul et al., 2014b, a, 2021; Subrahmanyam et al., 2005; Vinayachandran and Mathew, 2003; Zhang et al., 2016), especially for stronger, slowly moving TCs (Reul et al., 2021), with greater effects on the right-hand side of the storms in the Northern Hemisphere (Sun et al., 2021) and the left-hand side of the storms in the Southern Hemisphere. The TC-induced salty wake has also been described based on in-situ observations, generally for a single TC or for TCs over a specific region (Bond et al., 2011; Domingues et al., 2015; Sanford et al., 1987; Venkatesan et al., 2014; Zhang et al., 2018). A salty wake is consistent with TC-induced vertical mixing in regions where pre-event salinity increases with depth, i.e., as fresher water near the surface mixes with saltier water at depth.

A better understanding of freshwater redistribution in the upper ocean and its impacts on ocean stratification, during and after a weather event of interest, requires temperature and salinity measurements within the water column, such as those provided by Argo profiling floats. While Argo provides an unprecedented spatial and temporal coverage of the global ocean, most of the floats take measurements on a 10-day cycle (Roemmich et al., 2003, 2009), hence a composite approach of all the weather events of interest where Argo profiles are available is helpful (Lin et al., 2017). Using Argo observations and a composite approach, Steffen and Bourassa (2018) quantify barrier layer development due to tropical cyclones. A barrier layer is the salinity-stratified isothermal layer situated between the base of the mixed layer and the top of the thermocline (Godfrey and Lindstrom, 1989), in some regions of the ocean, and it acts as a barrier to the turbulent entrainment of cold thermocline water into the surface mixed layer (Cronin and McPhaden, 2002). Steffen and Bourassa (2018) show a statistically significant

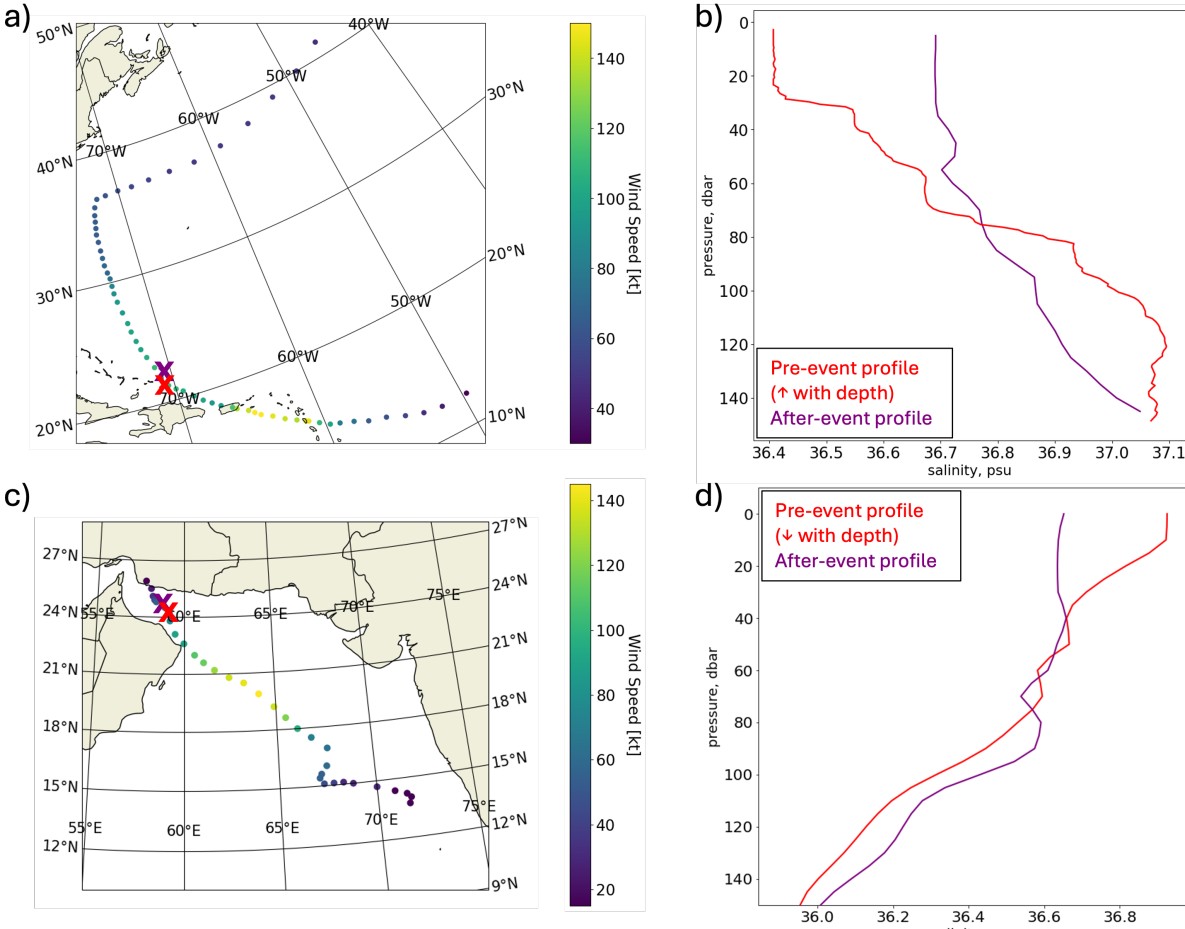

**Figure 1.** (a, c) Paths of two tropical cyclones and location of Argo profiles close in space and time to the TC tracks and collected before (red marker) and after (purple marker) the TC passage. (b, d) Comparison between salinity profiles collected before (red line) versus after (purple line) the TC passage. One of the pre-event salinity profiles increases with depth (panel b), the other decreases with depth (panel d).

increase in barrier layer thickness and barrier layer potential energy in the Atlantic basin with the TC-passage, consistent with an increase in the post-TC isothermal layer depth due to vertical mixing. On the other hand, the eastern Pacific basin shows no significant changes to any barrier layer characteristic, likely due to a shallow and highly stratified pycnocline, while the central Pacific has a statistically significant freshening in the upper 20–30 m, which increases upper-ocean stratification by about 35%. Tropical cyclones with different characteristics, such as intensity and forward speed, change the characteristics of the barrier layer (BL) thickness in different ways (Zhang et al., 2022): strong and slowly moving TCs have more impact on the upper ocean properties, with pronounced increase in the barrier layer's thickness. Finally, Balaguru et al. (2012) shows that the TC-intensification rate is almost 50% higher over regions with barrier layers, compared to regions without. The presence

of a preexisting ocean barrier layer can limit the effects of wind driven vertical mixing and the near-surface cooling response (Wang et al., 2011). This is the case, as while salinity increases from the bottom of the density based mixed layer to the bottom

of the barrier layer, temperature does not change much in the barrier layer, resulting in a more favorable ocean state for e.g., the maintenance of a tropical cyclone, as the mixed layer and thermocline are decoupled. A long-term freshening of the upper ocean also tends to intensify the strongest TCs of the western North Pacific, as the increase in stratification reduces their ability to cool the upper ocean (Balaguru et al., 2016).

In this work, we use Argo float observations (Argo, 2000; Roemmich et al., 2003, 2009) to study TC-induced changes in

upper ocean properties, focusing on hurricane-strength TCs, i.e., cyclones with maximum sustained winds greater or equal to 64 knots (32.9244 m/s). We describe these changes and their uncertainties using the method by Hu et al. (2024) and compare cases where the pre-event salinity profile increases versus decreases with depth, i.e., we compare two different initial conditions. Our goal is to describe the association between the "increasing" versus "decreasing" vertical structures of the pre-event salinity profile and changes in upper ocean salinity and density during hurricane-strength wind events, which has potential implications

for upper ocean stratification and air-sea exchanges during and after the event. If pre-event salinity increases with depth, wind-induced vertical mixing will result in an increase of near-surface salinity, as saltier waters from below are mixed in (Figure 1-b); if pre-event salinity decreases with depth, wind-induced vertical mixing will result in a decrease of near-surface salinity, as fresher waters from below are mixed in (Figure 1-d). For a given depth reached by the vertical mixing, the salinity change will be larger for pre-event salinity profiles with larger vertical gradients. The two different types of pre-event salinity profiles

(1-b, d; Figure S1-a in the Supporting Information S1) and associated near surface salinity changes cannot be captured when analyzing the presence versus absence of a pre-event barrier layer and the composite pre-event profiles with and without a barrier layer both show a vertical structure of salinity that increases with depth (Figure S1-b in the Supporting Information S1). Differences in near surface salinity changes for the "increasing" and "decreasing" case result in opposite contributions to the density changes with the weather event. As part of this study, we compare Argo-based results for hurricane-strength TCs with

the upper ocean response to hurricane-strength wind events in the HYCOM ocean reanalysis (Chassignet et al., 2007). The HYCOM reanalysis has been used in the past to investigate upper ocean physical and biological processes during hurricane-strength wind events (e.g., Gierach et al., 2009; Prasad and Hogan, 2007; Zamudio and Hogan, 2008) and complements our analysis as it provides time series for each event of interest, instead of sparse pairs of oceanic profiles before and after weather events, like in the case of Argo observations. We find that results from both Argo and HYCOM are consistent with the vertical

mixing of salinity playing a role in how upper ocean stratification changes with the TC passage. This is the case also for hurricane-strength wind events in general, as shown using HYCOM to investigate composites from only hurricane-strength wind events that are not co-located with observed tropical cyclones. Finally, we show that our results for hurricane-strength wind events do not change when we consider only pre-event "increasing" profiles with or without a barrier layer.

## 2   Data

### 2.1   Tropical cyclone track data

Tropical cyclone track data are available globally by combining products from the National Hurricane Center (NHC) and the United States Navy Joint Typhoon Warning Center (JTWC). The National Hurricane Center Data Archive (HURDAT2) database (Landsea and Franklin, 2013) includes TC track data for the North Atlantic and the Eastern Pacific basins. The JTWC database (Chu et al., 2002) provides TC track records for the Western Pacific Ocean, the Southern Hemisphere, and the North Indian Ocean.

Data are reported at six-hourly resolution, since 1851 for the Atlantic hurricane database (HURDAT2), since 1949 for the Northeast and North Central Pacific (HURDAT2) and since 1945 for the JTWC Best Track Data. In the following, we use TCs' spatio-temporal information and along-track wind speed during 2004–2020, i.e., during a time period when both TC track data and ocean temperature and salinity observations from Argo floats were available at the time of analysis.

### 2.2   HYCOM

The HYbrid Coordinate Ocean Model (HYCOM) reanalysis product is produced by the Naval Research Laboratory. In the following, we use output from the Global Ocean Forecasting System (GOFS) 3.1 (experiment 53.X), 41-layer HYCOM + Navy Coupled Ocean Data Assimilation (NCODA) Global 1/12° Reanalysis version of the product (Bleck, 2002; HYCOM Consortium, 2019). The vertical levels of HYCOM are different from those in other models as the vertical coordinates remain isopycnic in the open stratified ocean, smoothly transitioning to z-level coordinates in the weakly stratified upper-ocean mixed layer, to terrain-following sigma coordinates in shallow water regions, and back to z-level coordinates in very shallow water.

In our analysis, we focus on the time period $2011 - 2015$, which overlaps with Argo observations and when the model is forced by the National Centers for Environmental Prediction (NCEP) Climate Forecast System Version 2 (CFSv2; Saha et al., 2014). We use HYCOM temperature and salinity fields (available at 3-hourly temporal resolution), as well as the wind forcing (available hourly). From the HYCOM temperature and salinity fields, potential density is estimated using the TEOS-10 oceanographic toolbox (McDougall and Barker, 2011). Mixed layer properties are estimated using the methodology described in Holte and Talley (2009), i.e., a hybrid algorithm that models the general shape of each profile, searches for physical features in the profile, and calculates threshold and gradient MLDs to assemble a suite of possible MLD values, before selecting a final MLD estimate. This method has been shown to work well also in regions where the mixed layer exhibits great variability and the estimate of the MLD is overall challenging, e.g., winter mixed layers north of the Subantarctic Front, which can reach depths of 500 m and blend into deeper waters and remnant mixed layers (Holte and Talley, 2009).

## 2.3 Argo

Argo floats provide an unprecedented coverage in space and time of global ocean temperature and salinity fields in the upper 2,000 decibars (Argo, 2000; Roemmich et al., 2009), with extensions of the core array measuring ocean biogeochemical properties and at depths below 2,000 decibars.

In the following, we use Argo temperature and salinity profiles during 2004–2020, a time period that overlaps with available TC tracks at the time of analysis. Potential density, conservative temperature and absolute salinity are estimated from Argo-measured temperature, salinity and pressure using the TEOS-10 oceanographic toolbox (McDougall and Barker, 2011). As for HYCOM, mixed layer properties are estimated using the methodology described in Holte and Talley (2009).

## 3 Methods

### 3.1 HYCOM

We use the HYCOM output with high resolution in space and time to investigate upper ocean changes in the more general case of weather events with hurricane-strength winds (i.e., wind speed equal to or greater than 64 knots (32.9244 m/s)), before exploring upper ocean changes for the specific case of hurricane-strength TCs using sparse ocean observations. In the following we focus on 7 regions that include most of the events of interest between 32°S and 50°N (red boxes in Figure 2-a, b). For each region, we locate the events of interest starting from the maximum hurricane-strength wind event over the ocean and we exclude, from the remaining data, any other event within ±2° and 7 days before and after the one selected, to ensure selected events are independent. We then proceed in selecting subsequent events continuing with the strongest hurricane-strength wind event in the remaining wind data in the region: using this method, selected events are further than 7 days and ±2° from one another. We find that 32% of the 1238 events identified here are not co-located with an observed tropical cyclone, i.e., are not within a 200 km radius and +-1 day of an observed TC (henceforth referred to as non-TC events); most of these non-TC events are found in the North Atlantic and Pacific, with 35% in the North Atlantic box (in red in Figure 2-a,b) and 56% in the northernmost box in the Pacific (in red in Figure 2-a,b).

At the location of each selected hurricane-strength wind event, we store the time series of oceanic properties of interest, estimate the seasonal cycle, and remove it from the data to isolate the impact of hurricane-strength wind events on the upper ocean. To characterize this event-related signal, we analyze changes in oceanic properties (e.g., absolute salinity, and potential density) compared to 2 days before the event and focus on how the upper ocean evolves in the 14 days after the event. We use the profile 2 days before the event for the pre-event state, to hedge against storm effects prior to the event, consistent with Hu et al. (2024), Cheng et al. (2015), and with the method used for Argo in the next subsection.

We group selected events based on the pre-event vertical structure of salinity from the HYCOM reanalysis, i.e., the vertical structure 2 days before the event: "increasing" events are located where pre-event absolute salinity increases between the density based mixed layer and 50m below the mixed layer (e.g., Figure 1-b); "decreasing" events, where salinity decreases between the mixed layer and 50m below (e.g., Figure 1-d). The number of "increasing" events is much larger than the number

of "decreasing events", with the latter mostly located in the Northern Hemisphere (Figure 2). We use 50 meters for the thickness of the layer considered below the mixed-layer, as it captures the 90th percentile (across events) of the observed mixed layer deepening (not shown). We find our conclusions in the following do not change with the thickness of the layer considered below the mixed-layer, e.g., if we consider 20m to 70m below the mixed-layer.

Composites of seasonally-adjusted differences between profiles at each time step and the profile 2 days before the event are created for the "increasing" and "decreasing" group and the two groups are compared, see Figure 3, 4. For each composite, we compute the standard error and use it to identify significant signal (95% confidence level). As the two groups of events are independent by construction, the variance of the difference (between "increasing" and "decreasing") is estimated as sum of the variances.

## 3.2 Argo

Our analysis of TC-related upper ocean changes from (sparse) Argo observations is based on the differences between post-TC and pre-TC Argo profiles, as described in Hu et al. (2024). Hu et al. (2024) introduced a framework that, for the first time, 1. accounts for seasonal effects to isolate the signal of interest, 2. models space-time covariances to provide rigorous uncertainty quantification for results from sparse observations, and 3. treats time as a continuous variable (rather than producing estimates that are binned in time, as other studies). In the following, we provide a summary of the method in Hu et al. (2024), and include a description of differences in our implementation.

### 3.2.1 Defining a TC-centric coordinate system

Given the scattered nature, in space and time, of the Argo data, we define a TC-centric coordinate system to characterize the ocean response to TCs' passage in the continuous time and space realms (Hu et al., 2024). One coordinate is the time difference between the post-TC Argo profile and the TC passage. The other coordinate is the cross-track angle, which is determined by calculating the angle between line segments from the sphere's center to two surface points, equivalent to the haversine distance. When longitude is constant, this angle is the absolute difference in latitude. When latitude is constant, it is the absolute difference in longitude, adjusted by the cosine of the latitude. The cross-track angle aligns with the great circle distance and is used here as it provides a more accurate spherical model for measuring the distance from the TC track, compared to e.g., using distance in km.

### 3.2.2 Selecting Argo profile pairs to characterize changes with TC passage

To investigate TC-related upper ocean changes, we select "TC profiles" that are within 12 days before and 20 days after a TC event and within a cross-track angle of $\pm 2°$ of a TC event, i.e., $\pm 2°$ of a TC track data point; as indicated earlier, we only consider TC track data points that correspond to hurricane-strength TCs. All the other Argo profiles, which represent the vast majority, are classified as "non-TC profiles".

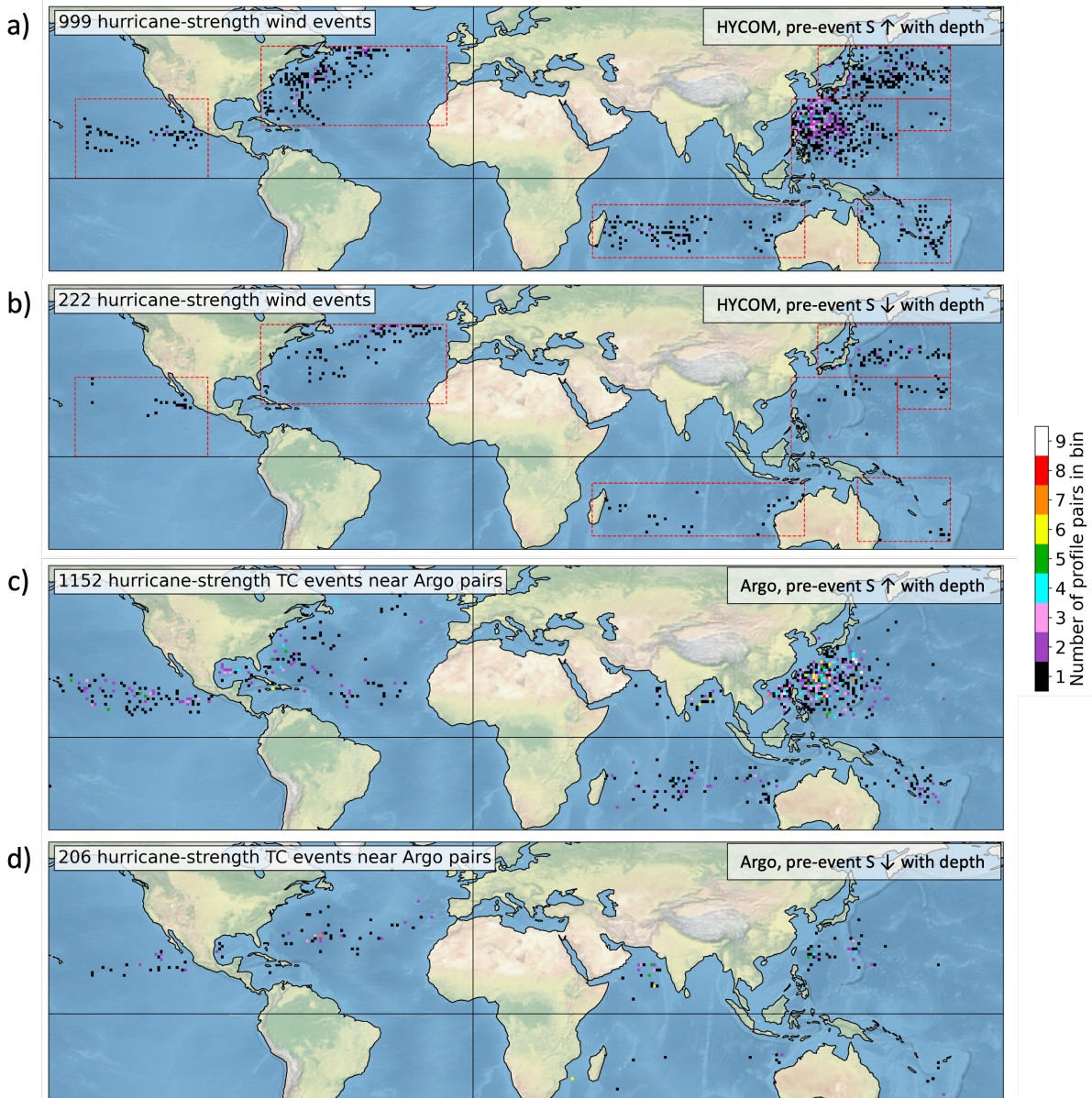

**Figure 2.** (a, b) Number of weather events included in the HYCOM analysis, in $1 \times 1$ degree bins, i.e., weather events with hurricane-strength winds during 2011–2015 and within the regions outlined in red. (c, d) Number of Argo profile pairs included in the Argo analysis, in $1 \times 1$ degree bins, i.e., Argo profile pairs colocated with hurricane-strength TCs during 2004–2020. Panels a and c are for locations where pre-event upper ocean salinity increases with depth. Panels b and d are for locations where pre-event upper ocean salinity decreases with depth.

TC profiles are used to create pairs of Argo profiles: each pair includes one profile before a TC, and one profile after, and their difference is used to assess the ocean response to TCs. To be paired, two Argo profiles need to be both incidental to a TC

track (within $\pm 2°$ in cross-track angle) and colocated in space (within $\pm 0.2°$) and with a time separation of 32 days or less. As in Hu et al. (2024), the pre-TC Argo profile, which we can also call the "baseline" profile, needs to be between 12 days and 2 days before the TC passage; the post-TC Argo profile, which we also call the "signal" profile, needs to be between 2 days before and 20 days after the TC passage to hedge against storm effects prior to the event, as described Hu et al. (2024) and consistent with Cheng et al. (2015).

### 3.2.3 Classification of profiles based on the pre-event vertical structure of salinity

As for HYCOM and differently from Hu et al. (2024), we group Argo profile pairs based on the vertical structure of the Argo salinity profile before the weather event: "increasing" pairs include Argo pre-TC profiles with upper ocean salinity increasing with depth (e.g., Figure 1-b); "decreasing" pairs include Argo pre-TC profiles with salinity decreasing with depth (e.g., Figure 1-d). For each of the two groups, we estimate upper ocean changes with the TC passage and compare them to one another to characterize differences associated with the pre-event vertical structure of salinity. As for HYCOM, there are many more instances for the "increasing" case compared to the "decreasing case" (Figure 2-c, d). We note that, while the total counts in Figure 2 (white box on the top left of each map) are comparable between HYCOM and Argo, the numbers have different meanings. For HYCOM, the number indicates the count of all selected weather events, hence the number of continuous time series of ocean temperature/salinity available for the analysis, as the model output is available at the location of the weather event at all times of the model simulation. For Argo, the number indicates the count of all available profile pairs to estimate the upper ocean response to the TC passage, as we only have sparse observations from Argo floats.

### 3.2.4 Estimating and removing effects of the seasonal cycle

To isolate the effect of hurricane-strength TCs on upper ocean properties based on differences between post-event and pre-event profiles, the seasonal cycle needs to be removed from the data. For example, as observed in Hu et al. (2024), the TC season is usually associated with a seasonal warming, which therefore needs to be removed from the data to study TC-related upper ocean changes. As in Hu et al. (2024), we estimate the seasonal cycle using the local linear regression method in Roemmich and Gilson (2009) and all the Argo profiles that were classified as non-TC. First, we calculate the local linear regression on a 1x1 degree grid; then, each TC profile is matched to the closest grid point to remove the seasonal cycle from that profile, i.e., we subtract the seasonal mean field estimate from the pre-TC and post-TC Argo profiles to then analyze seasonally-adjusted differences.

### 3.2.5 Determining the statistically significant, smoothed results

In addition to the TC-related signal of interest, seasonally-adjusted variables from TC-profiles include effects of ocean variability that is unrelated to TCs, i.e., variability that represents noise in our analysis. As in Hu et al. (2024), we characterize this variability by fitting the Gaussian process model described in Kuusela and Stein (2018) to non-TC Argo profiles. Once fitted, the model is used to provide information about ocean variability at the time and locations of the TC profiles and estimate

self-variances and cross-covariances of seasonally-adjusted variables. As in Hu et al. (2024), only the non-TC activity is represented here as a random process, hence, the variance of the before and after profiles is the same under this model. The estimate of self-variances and cross-covariances of seasonally-adjusted variables allows us to account for cross-observation dependence

and point-wise variance when smoothing seasonally-adjusted differences between the post-TC profile and the pre-TC profile, to characterize the upper ocean response to the passage of hurricane-strength TCs as a continuous function of cross-track angle and time, for different vertical levels. Notably, this allows us to obtain point-wise confidence intervals on our estimates ($\alpha$=0.05, i.e., to obtain 95% coverage), hence to assess the significance of each individual pixel in our plots, as shown in Figure 5, 6. We smooth seasonally adjusted differences using a fixed-knot thin plate spline smoother (Duchon, 1977; Green and Silverman,

1993; Nychka, 2000; Wahba, 1980, 1990; Wood, 2017).

## 4 Results and discussion

### 4.1 Upper ocean changes during hurricane-strength wind events in HYCOM

Consistent with previous studies (Black and Dickey, 2008; Han et al., 2024; Hu et al., 2024; Korty et al., 2008; Price, 1981; Zhang et al., 2021), the analysis of HYCOM fields shows that, during hurricane-strength wind events, upper ocean temperature

decreases (Figure S2-a, b in Supporting Information S1). Several factors can however impact the magnitude of such change. As an example, the presence or absence of a strong barrier layer can significantly influence the heat exchange processes at the air-sea interface and affect e.g., the intensity and track of tropical cyclones (Balaguru et al., 2012; Reul et al., 2021; Wang et al., 2011). A strong barrier layer reduces the cooling of the upper ocean during the passage of a TC (e.g., Reul et al., 2021), as it can limit the vertical mixing of cold waters from below the mixed layer, allowing the storm to maintain and even intensify

its strength (Balaguru et al., 2012). Also, with a pre-event barrier layer, when vertical mixing reaches below the density-based mixed layer, it may only mix in waters with similar temperature as temperature within the barrier layer is similar to the one above by definition. On the other hand, in the absence of a pre-event barrier layer, the upper ocean cooling with the TC passage is larger, which may weaken the storm. While not the focus of our analysis, the comparison of HYCOM temperature composites based on the presence or absence of a barrier layer before a hurricane-strength wind event yields results that are consistent with

previous TC studies, i.e., the cooling after the event is overall larger in the absence of a barrier layer (not shown). A difference in cooling is also observed when comparing HYCOM composites based on the vertical structure of salinity before the event, i. e., pre-event salinity "increasing" versus "decreasing" (instead of pre-event barrier layer present versus not). In particular, the cooling is larger for the "increasing" case (Figure S2-a in Supporting Information S1) than for the "decreasing" case (Figure S2-b in Supporting Information S1, Figure 4-c). The "increasing" case includes most of the events, with $\approx$30% of these events

in regions with a barrier layer. If we compare "increasing" vs "decreasing" composites for only events with no pre-event barrier layer, we still find a larger cooling of the top $\approx$ 15m for the "increasing" case in the days after the event (Figure S3-d, e, f in Supporting Information S1), yet, the difference is much smaller and limited to day 2 to 5 after the event (Figure S2-f in Supporting Information S1). Also, upper ocean temperature differences between the "increasing" and "decreasing" case are not significant when only considering non-TC hurricane-strength wind events (Figure S4-d, e, f in Supporting Information

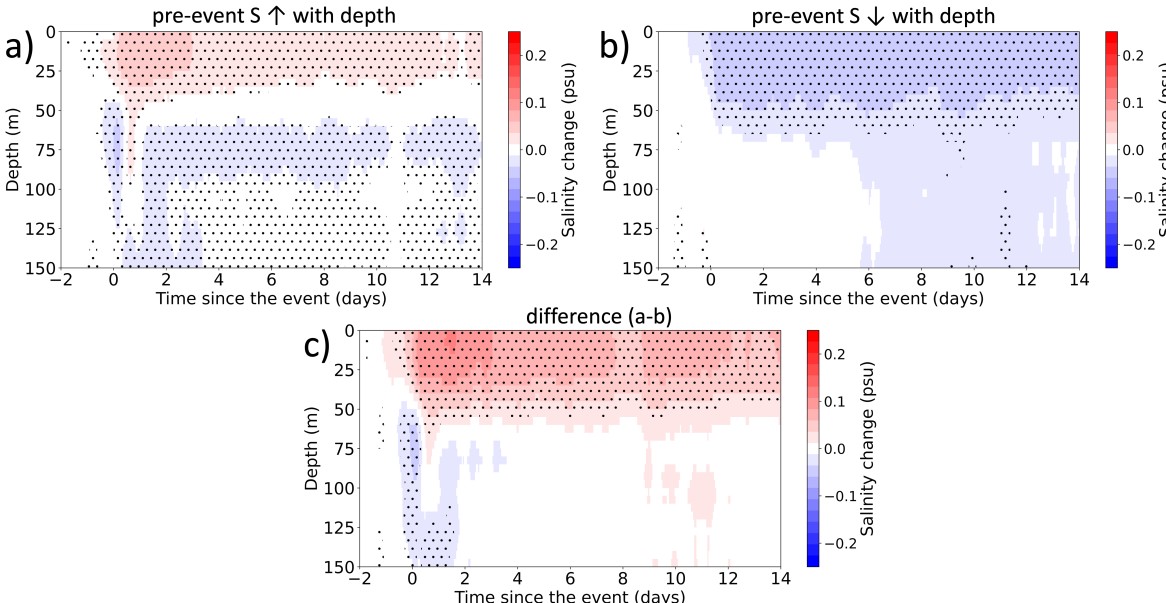

**Figure 3.** Upper ocean salinity changes during hurricane-strength wind events, based on the HYCOM reanalysis. Composite changes are shown for regions where pre-event upper ocean salinity increases (a) vs decreases (b) with depth. Panel (c) shows the difference between the "increasing" case (i.e., panel a) and the "decreasing" case (i.e., panel b). Dots indicate statistically significant values (95% confidence limit).

S1). We note that factors such as wind strength and duration of the strongest winds are key to understand the upper ocean temperature response to weather events, i.e., storm intensity, size and translation speed affect the characteristics of the upper ocean temperature response to the event (Anthes and Chang, 1978; Emanuel and Nolan, 2004; Lin et al., 2017; Samson et al., 2014; Wang et al., 2016; Zhu and Zhang, 2006). Isolating the difference in cooling between the "increasing" and "decreasing" case from other relevant factors may be difficult with a limited number of events, hence it is not a focus here. The discussion of temperature is included as relevant to better understand results for density presented later in this section.

Consistent with wind-induced vertical mixing, near surface salinity increases in regions where pre-event salinity increases with depth, due to fresher waters mixing with saltier waters below (Figure 3-a); vice versa, near surface salinity decreases where pre-event salinity decreases with depth, as saltier waters mix with fresher water below (Figure 3-b). More specifically, for the "increasing" composite, ocean salinity increases by over 0.05 psu in the top 40 m after hurricane-strength wind events (Figure 3-a). This positive anomaly weakens in time, yet is still detectable after 14 days. For the "decreasing" composite, a decrease by over 0.05 psu is seen instead (Figure 3-b). The difference between the two composites is statistically significant (Figure 3-c). This is also seen when only locations with no pre-event barrier layer are considered (Figure S3-a, b, c in Supporting Information S1). Additionally, the same result is found when only non-TC hurricane-strength wind events are included in the composites (Figure S4-a, b, c in Supporting Information S1). For the "increasing" composite, salinity decreases between 50 m and 100 m, yet the signal is weaker compared to the increase near the surface: this difference in sign of salinity changes at

different depths is consistent with wind-induced vertical mixing acting on regions with a pre-event salinity that increases with depth, i.e., near surface waters get saltier as they mix with saltier waters at depth; deeper layers get fresher as they mix with lower salinity waters above. Yet, while an increase in salinity near the surface is common to most profiles in the composite, where the decrease happens in the water column may depend on the details of the stratification at different locations and a switch in sign of the observed changes with depth may not have appeared in the composite, e.g., for the "decreasing" case, no significant salinity increase is detected in the top 150 m (Figure 3-b). As for temperature, results for salinity in this study are consistent with previous research reporting sea surface salinity (SSS) salinification in the trail of TCs (e.g., Bond et al., 2011; Chaudhuri et al., 2019; Domingues et al., 2015; Grodsky et al., 2012; Lin et al., 2017; Liu et al., 2020; McPhaden et al., 2009; Price, 1981; Reul et al., 2014b; Sanford et al., 1987; Steffen and Bourassa, 2018; Venkatesan et al., 2014; Vinayachandran and Mathew, 2003; Zhang et al., 2016, 2018). This SSS salinification, observed from satellites particularly for slow-moving TCs, is consistent with the importance of vertical mixing during hurricane-strength wind events and the fact that subsurface water is on average saltier than surface water in convective regions associated with TC activity (Jourdain et al., 2013); hence, not differentiating between the "increasing" versus "decreasing" case as in the present study, will yield a result closer to the "increasing" case. Also, as described in Reul et al. (2021) and found here in HYCOM composites (not shown), barrier layers lead to storm wakes that are saltier compared to wakes over barrier layer free areas.

Finally, consistent with the described cooling, for both "increasing" and "decreasing" composites, upper ocean potential density increases with the passage of hurricane-strength wind events (Figure 4-a, b). The increase in density is larger for the "increasing" case, as both the cooling is stronger and upper ocean salinity increases, and the difference with the "decreasing" case is statistically significant (Figure 4-c). While both temperature and salinity contribute to the stronger increase in density for the "increasing" case, an estimate of the temperature versus salinity contribution to the difference between the "increasing" versus "decreasing" composites for density indicates salinity contributes around $35-40\%$ (not shown). As for temperature and salinity, similar results are found for the difference between "increasing" versus "decreasing" composites for density when only locations with no pre-event barrier layer are considered (Figure S3-g, h, i in Supporting Information S1). This is the case also when only non-TC hurricane-strength wind events are included in the composites (Figure S4-a, b, c in Supporting Information S1). While results here are overall consistent with previous studies, our analysis shows the importance of an "increasing" versus "decreasing" vertical structure of pre-event salinity for how density changes in the upper ocean with hurricane-strength wind events, which can modulate air-sea interactions during and after the events through changes in stratification.

## 4.2   Upper ocean changes observed by Argo floats during hurricane-strength tropical cyclones

While HYCOM assimilates observations to provide a model representation of relevant processes at high resolution in space and time, the question remains of how results in Section 4.1 compare with in-situ measurements during relevant weather events. In the following, we discuss differences and similarities between the Argo- versus HYCOM-based results, and refer to the previous section for some of the context from previous studies.

Consistent with results in Section 4.1 and several previous studies (including analyses based on Argo data, e.g., Liu et al., 2007; Cheng et al., 2015), Argo observations show upper ocean cooling as hurricane-strength TCs move over the ocean, for

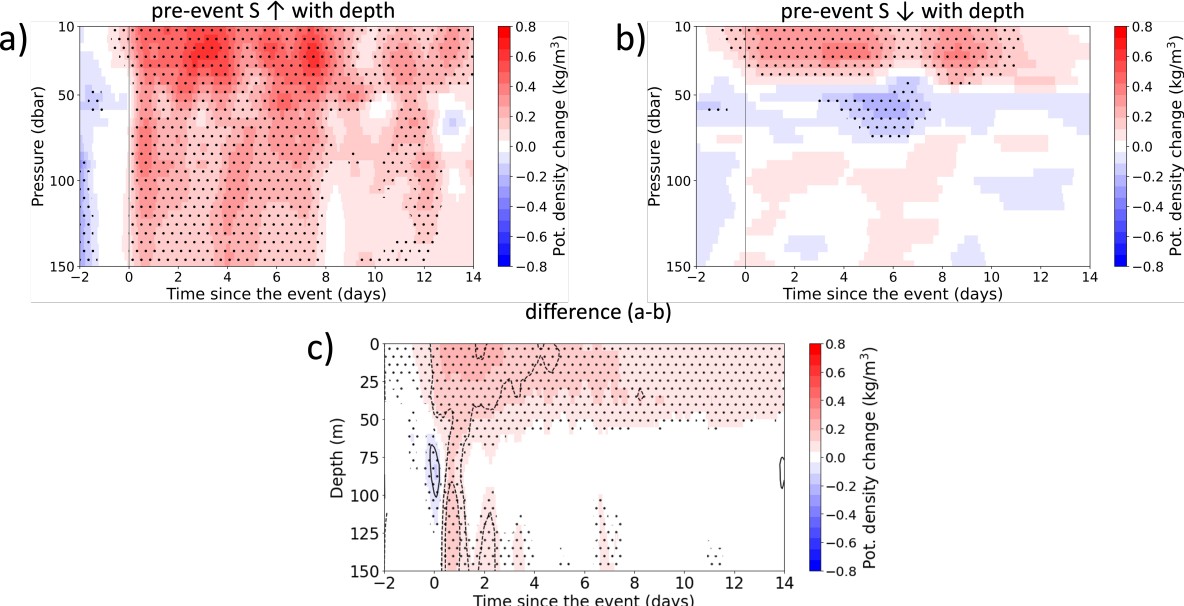

**Figure 4.** As in Figure 3, now for potential density. Black contours in panel (c) indicate differences in temperature change between the "increasing" and "decreasing" case and are shown only when significant, as in Figure S2-c in Supporting Information S1. A continuous line indicates positive values, a dashed line indicates negative values.

both the "increasing" and "decreasing" case (Figure S2-a in Supporting Information S1). Different from HYCOM, observed cooling for the "increasing" case in Argo is detected as deep as 150 dbar for the 2 weeks after the event (Figure S5-a in Supporting Information S1), i.e., deeper than what seen in HYCOM fields for hurricane-strength wind events. This difference may be related to e.g., how vertical mixing processes are represented in the HYCOM model, and in particular how deep the wind-induced mixing reaches with the event; the details of the vertical structure of pre-event upper ocean properties in the
model versus observations, i.e., the initial condition of events of interest; the availability of sparse ocean observations co-located with events of interest compared to the continuous time series the HYCOM model provides in its spatial domain. Also, for the "decreasing" case in Argo, a warming is observed between 50 and 70 dbar (Figure S5-b in Supporting Information S1). Analogous to what was discussed for Figure 3-a, vertical mixing of the water column may result in the observed warming, as warmer waters above are mixed downward. However, it is not surprising that the same vertical pattern of changes, with
different signs at different depths, is not seen in Figure S5-a in Supporting Information S1. While the near surface cooling is common across locations used to estimate ocean changes for the "increasing" versus "decreasing" composite, the depth of the warming signal and its vertical extent vary, (analogous to what discussed for Figure 3-a). The challenges described in section 4.1 to characterize the differences in temperature changes between the "increasing" and "decreasing" case apply also to results from sparse Argo observations. While the cooling below 50 dbar is stronger for the "increasing" case compared to "decreasing",
the cooling in the top 50 dbar is stronger for the "decreasing" case or not statistically different between the two groups (Figure

6 and Figure S5-c in Supporting Information S1). As for HYCOM, differences in temperature changes between "increasing" and "decreasing" case are not the focus of our analysis, yet Figure 6 and Figure S5-c in Supporting Information S1 are helpful to interpret differences in density changes later in this section.

Upper ocean salinity and density changes observed by Argo during hurricane-strength TCs are overall consistent with
325 HYCOM-based results (Section 4.1). Consistent with wind-induced vertical mixing, salinity increases with the TC passage for the "increasing" case, and it decreases for the "decreasing" case (Figure 5). Hence, salinity enhances or reduces the upper ocean increase in density that occurs as the upper ocean cools with the TC passage (Figure 6). Even if overall changes in salinity and density are consistent between the Argo and HYCOM analyses, some of the details may be different. As an example, the vertical structure of Argo-based density changes for the "decreasing" case reflects the vertical structure of changes in
temperature described earlier in this subsection, i.e., a switch in sign between shallower and deeper layers, hence it is different from HYCOM. Also, different from what discussed for HYCOM at the end of the previous subsection, in Argo, the differences in upper ocean density changes between the "increasing" and "decreasing" case largely reflect differences in salinity changes (Figure 5-c, top 50 dbar), rather than both differences in temperature changes (Figure S5-c in Supporting Information S1) and salinity changes: the stronger increase in density for the "increasing" case is associated with salinity "increasing" where pre-
event salinity increases with depth, as wind-induced mixing brings up saltier water, and is observed despite a weaker cooling for the "increasing" case compared to the "decreasing" case.

Finally, Argo-based changes in mixed layer properties with the TC passage largely reflect what was described earlier for changes at different vertical levels and in time. Looking at these changes as a function of the cross-track angle (e.g., Figure 7 for the "increasing" case), shows the asymmetric signature of wind-driven vertical mixing with the TC passage; a stronger signal
is observed on the right-hand side of the storms in the Northern Hemisphere, and on the left-hand side of the storms in the Southern Hemisphere, e.g., documented for salinity in Sun et al. (2021). We note that, while panels in Figure 7 include events in both the Northern and Southern Hemisphere, we flip the sign of the cross-track angle for TCs in the Southern Hemisphere when estimating changes in ocean properties, in order to display the asymmetry in a consistent manner across hemispheres.

## 5  Summary and conclusions

In this study, we characterize observed upper ocean changes during hurricane-strength tropical cyclones. We focus on 1. the statistical significance of the changes associated with different pre-event vertical structures of salinity, i.e."increasing" versus "decreasing" with depth, and 2. the contribution of salinity changes to upper ocean density changes in the "increasing" versus "decreasing" case. Upper ocean salinity increases with the TC passage in "increasing" locations, i.e., the majority of the areas with cyclonic activity; vice versa, it decreases in "decreasing" locations. This observed difference is consistent with wind-
induced vertical mixing of the upper ocean with the TC passage, e.g., for the "increasing" case, fresher near-surface waters mix with saltier waters below. The difference in salinity changes for the "increasing" versus "decreasing" case results in differences in how upper ocean density changes: while near surface density increases in both the "increasing" and "decreasing" case, consistent with cooling due to air-sea exchanges of heat and vertical mixing, it increases more for the "increasing" case, due

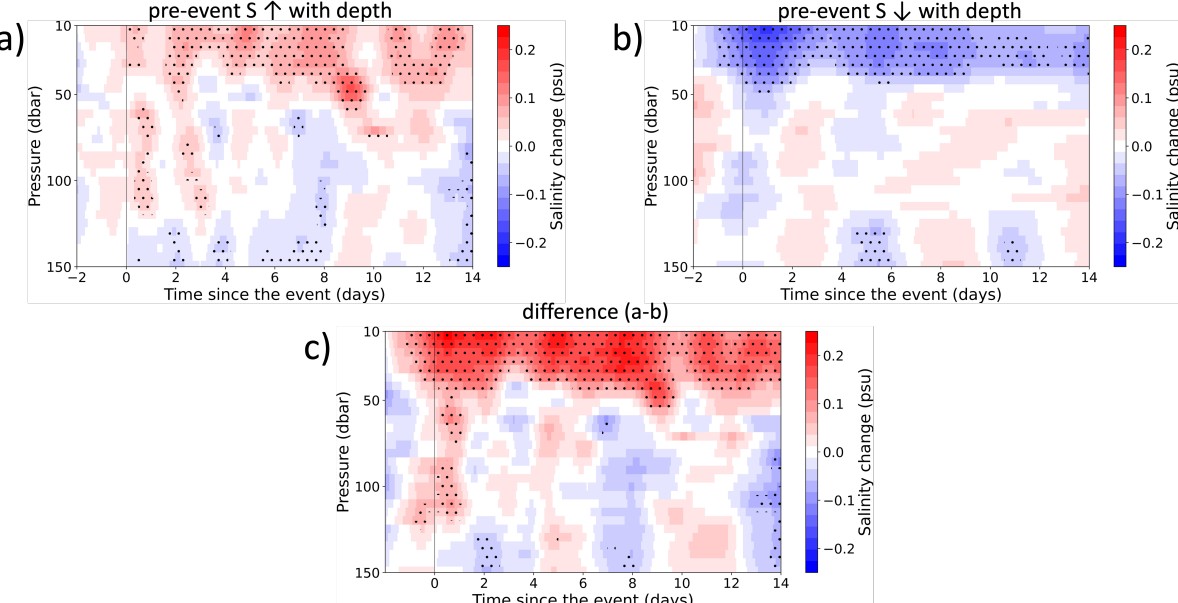

**Figure 5.** Upper ocean salinity changes within 0.5 degrees in cross track angle from the TC track during hurricane-strength TCs, based on Argo observations. Changes are shown for regions where pre-event upper ocean salinity increases (a) vs decreases (b) with depth. Panel (c) shows the difference between the "increasing" case (i.e., panel a) and the "decreasing" case (i.e., panel b). In all panels, a point-wise $\alpha = 0.05$ hypothesis test is performed and used to indicate (with dots) where the null hypothesis is rejected.

to the contribution from the salinity increase, with potential changes in stratification, i.e., as density increases with depth in a
stable water column, a larger increase in near-surface density is consistent with a larger decrease in stratification. Results from Argo observations are discussed in the more general context of upper ocean changes during hurricane-strength wind events (not just tropical cyclones) using the HYCOM reanalysis. Results from Argo and HYCOM are overall consistent with one another and complement each other, as while HYCOM provides high resolution fields in space and time it may be limited in representing all relevant processes and in capturing the details of the pre-event vertical structure of upper ocean properties in
the real ocean.

Our focus on the association between the pre-event vertical structure of salinity, "increasing" versus "decreasing" with depth, and upper ocean salinity and density changes with the TC passage adds to the existing literature on air-sea interactions during TCs. Changes in the vertical structure of upper ocean density may have implications for air-sea interactions during the weather event and after, hence a better understanding of relevant processes has the potential to improve the prediction of TC
intensification and, more in general, the model representation of air-sea interactions during the season when hurricane-strength wind events occur. As an example, insights from our work could inform process model experiments to study air-sea interactions in "increasing" versus "decreasing" regions during and after hurricane-strength wind events, on the lines of e.g., Balaguru et al. (2012); Hoffman et al. (2022); Iyer and Drushka (2021); (not all focusing on TCs).

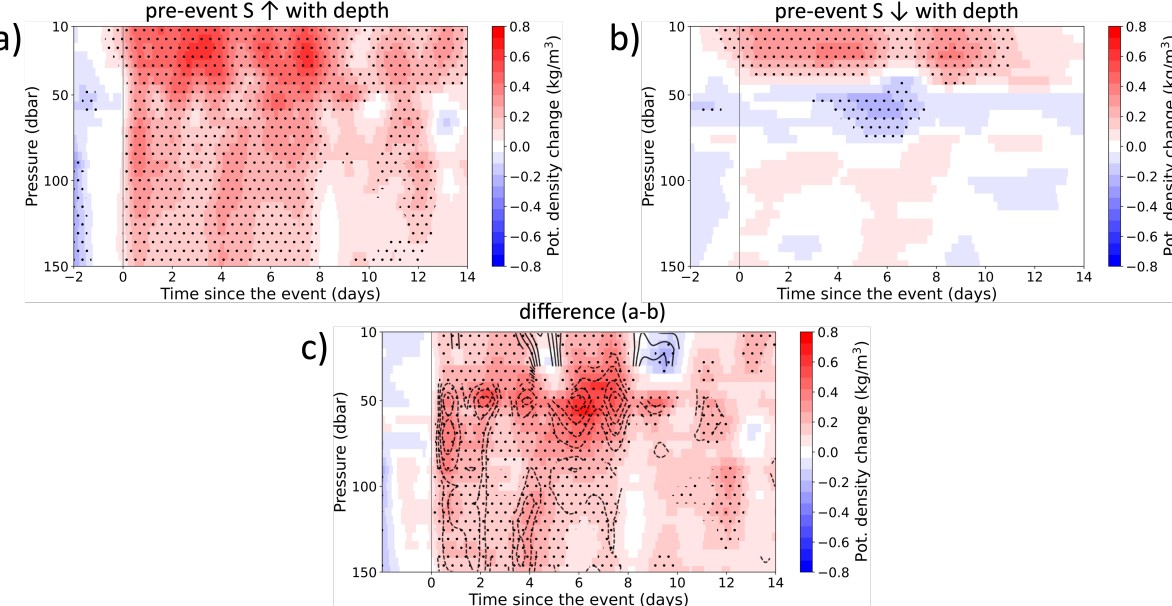

**Figure 6.** As in Figure 5, now for potential density. Black contours in panel (c) indicate differences in temperature change between the "increasing" and "decreasing" case and are shown only when significant, as in Figure S5-c in Supporting Information S1. A continuous line indicates positive values, a dashed line indicates negative values.

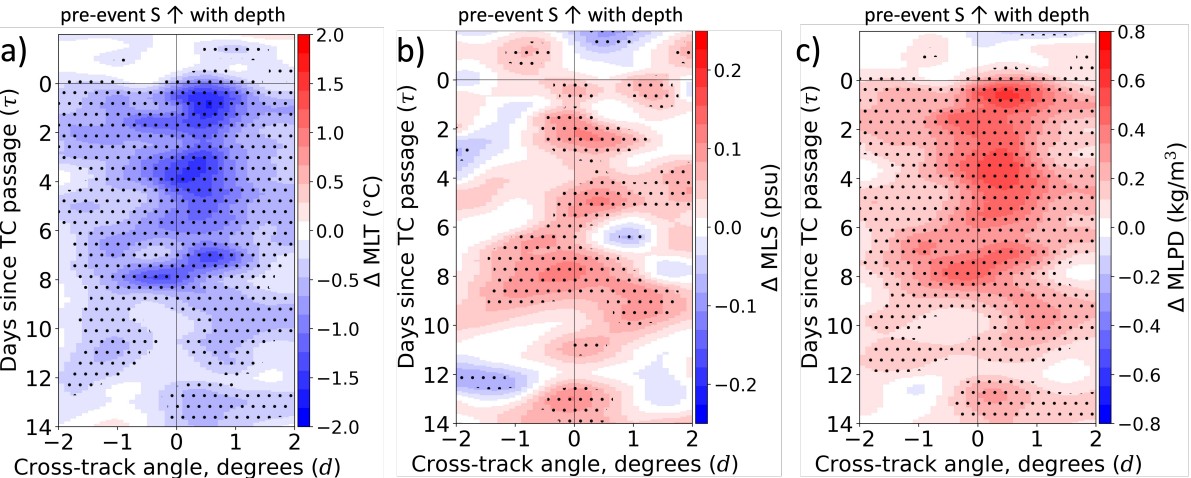

**Figure 7.** Upper ocean changes in mixed layer temperature (MLT, panel a), salinity (MLS, b) and potential density (MLPD, c) during hurricane-strength tropical cyclones, based on Argo observations. Changes are shown in regions where pre-TC salinity increases with depth. In all panels, a point-wise $\alpha = 0.05$ hypothesis test is performed and used to indicate (with dots) where the null hypothesis is rejected.

Several opportunities also exist to extend our work in other directions, including for other weather events of interest or using satellite observations. As one example, changes in the ocean biogeochemistry during hurricane-strength TCs or other weather events could be investigated using our framework, as the fleet of biogeochemical Argo floats grows in time, e.g., changes in chlorophyll-a, oxygen, nitrate, pH (Bittig et al., 2019). Finally, our framework can be leveraged to compare how different reanalysis products represent air-sea exchanges during weather events in the context of the insights from sparse observations.

*Data availability.* Argo data are made freely available by the International Argo Program and the national programs that contribute to it (http://argo.jcommops.org). Tropical cyclone track data are made available by NOAA (https://www.nhc.noaa.gov/data/#hurdat) and the Joint Typhoon Warning Center (https://www.metoc.navy.mil/jtwc/jtwc.html?best-tracks). HYCOM data are available at https://www.hycom.org/data/glbv0pt08/expt-53ptx. Argo profiles and tropical cyclone track data were also accessed via Argovis (https://argovis.colorado.edu, https://argovis.colorado.edu/argo, https://github.com/argovis/demo_notebooks).

*Author contributions.* All authors contributed to the conceptualization and design of this study. JS conducted the analysis and wrote the manuscript, with contributions from DG, AH, and MK. The final manuscript underwent a thorough review and editing process, led by JS, DG, AH, MK and KMW, ensuring its quality and accuracy.

*Competing interests.* The author has declared that none of the authors has any competing interests.

*Acknowledgements.* Jacopo Sala and Donata Giglio acknowledge support by NSF award 1928305, NASA award 80NSSC19K0059, and NOAA award NA21OAR4310261. Addison J. Hu acknowledges support from the NSF GRFP (Award DGE175016) and NSF DMS (Award 1520786). Mikael Kuusela acknowledges support from NOAA award NA21OAR4310258. We would also like to acknowledge high-performance computing support from the supercomputer Cheyenne provided by the National Center for Atmospheric Research's (NCAR) Computational and Information Systems Laboratory, sponsored by the National Science Foundation (NSF). These data were collected and made freely available by the International Argo Program and the national programs that contribute to it. (https://argo.ucsd.edu, https://www.ocean-ops.org). The Argo Program is part of the Global Ocean Observing System. Funding for the development of HYCOM has been provided by the National Ocean Partnership Program and the Office of Naval Research. Data assimilative products using HYCOM are funded by the U.S. Navy. Computer time was made available by the DoD High Performance Computing Modernization Program. The output is publicly available at https://hycom.org.

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
