# Peer review of "Upper ocean changes with hurricane-strength wind events: a study using Argo profiles and an ocean reanalysis"

_EGUsphere, 2024_

## Author Response (AR1)

**Review 1**

The authors would like to thank the reviewer for their thoughtful comments and suggestions. Our responses are below (reviewer's comments are indicated with black bold font).

**The authors examine how pre-existing salinity stratification impacts the vertical redistribution of salinity by TCs and weather events with TC-strength winds. A strength of the paper is the use of both HYCOM ocean reanalysis fields as well as in situ measurements by Argo profiling floats. The authors find that pre-event salinity profiles that increase with depth yield greater near-surface salinity increases and near-surface stratification decreases compared to pre-event salinity profiles that decrease with depth. The analysis and results are potentially interesting and informative to the TC and air-sea interaction communities, but several improvements to the writing and presentation are needed to make the paper suitable for publication.**

**First, the writing is rather negatively affected by the overuse of parenthetical statements, making it difficult to follow the logic and thought processes of the authors. Examples just within the Introduction can be found on lines 38, 46, 56, 57, 63, and 72, but this practice is pervasive throughout the entire manuscript. In general, text within parentheses should either be omitted, as it is not essential to convey the concept or finding, or should be fully incorporated into the sentence; the latter may require sentence restructuring. There are also several instances of run-on sentences (i.e., lines 60-63) that require attention. Additionally, the use of "condition (counter-condition) to describe state (counter-state)" is discouraged in scientific writing. See the 2010 Eos article by Alan Robock "Parentheses Are (Are Not) for References and Clarification (Saving Space)" for a discussion on this point.**

Thank you for your suggestions to improve the writing and presentation. We made edits throughout the manuscript to 1. limit the use of parenthetical statements and 2. eliminate run-on sentences and condition/counter-condition sentences. Some examples of updated sentences are listed in the following:

- (Line 7 in the version of the pdf with tracked changes): "Results show that in regions where pre-event salinity increases with depth there is a corresponding statistically significant increase in upper ocean salinity; vice versa, we observe a significant decrease in upper ocean salinity in regions where pre-event salinity decreases with depth."
- (Line 42 in the version of the pdf with tracked changes): "This limits our understanding of how these changes can impact air-sea interactions during and after the TC passage, as the degree of stratification of the water column is related to vertical gradients of temperature and salinity and affects the strength of vertical mixing induced by atmospheric disturbances, including hurricanes."

- (Line 52 in the version of the pdf with tracked changes): "The TC-induced salty wake has also been described based on in-situ observations, generally for a single TC or for TCs over a specific region (Bond et al., 2011; Domingues et al., 2015; Sanford et al., 1987; Venkatesan et al., 2014; Zhang et al., 2018); a salty wake is consistent with TC-induced vertical mixing in regions where pre-event salinity increases with depth, i.e. in most regions where cyclonic activity is found, as fresher water near the surface mixes with saltier water at depth."
- (Line 66 in the version of the pdf with tracked changes): "On the other hand, the eastern Pacific basin shows no significant changes to any barrier layer characteristic, likely due to a shallow and highly stratified pycnocline. Also, the central Pacific has a statistically significant freshening in the upper 20–30 m, which increases upper-ocean stratification by ~35%."
- (Line 76 in the version of the pdf with tracked changes): "A long-term freshening of the upper ocean also tends to intensify the strongest TCs of the western North Pacific, as the increase in stratification reduces their ability to cool the upper ocean (Balaguru et al. 2016). "

More examples can be found at lines 166, 175, 200, 211, 273, 363, 378 in the version of the pdf with tracked changes.

**Other points to address before publication relate to the motivation and framing of the paper, analysis methods, the use of supplemental figures to discuss temperature changes, and figure improvements.**

Thank you for your comments. Each was appreciated and led to new text/figures that are described in the following.

**Motivation/framing: I could not understand the rationale to analyze TC effects for increasing and decreasing salinity profiles instead of barrier layer presence, and how this classification might be similar to or different than previous work that has focused on barrier layers. To make a case for their approach, the authors should 1) show mean S or S' profiles for each state and 2) include an analysis of barrier layer presence, and possibly thickness and strength (i.e., stability) as a function of salinity profile.**

Thank you for your suggestions. To clarify the rationale for our analysis we included:

- New text and a new figure in the "Introduction":
    - The figure shows an example pre-event salinity profile for each of the two groups, i.e. for pre-event salinity increasing versus decreasing with depth.
    - The text refers to the figure to clarify the rationale for our analysis and it clarifies that we also analyze the effect of the presence of a barrier layer on our results (when a barrier layer is present, salinity increases from the bottom of the density based mixed layer to the bottom of the barrier layer). We show that our results do not change regardless of the presence or absence of a pre-event barrier layer.

- A new figure in the Supporting Information S1 showing 1. the composite pre-event profile for each of the two groups, i.e. for pre-event salinity increasing versus decreasing with depth; 2. how composite pre-event profiles with and without barrier layers are characterized by salinity increasing with depth.
- Edits in the "Methods" to refer to the new figures.

New text and figures can be found in the following. Figure numbers refer to the updated set of figures. Please note that a lighter gray font indicates text in the original submission (even if edits were made in response to other reviewer's comments) and is included here for context:

→ New text in the "Introduction" to clarify the rationale for our analysis, as well as clarify that we also analyze the effect of the presence of a barrier layer on our results (the new text will appear in the revised manuscript starting at line 60 of the original pdf):

Also, Balaguru et al. (2012) shows that the TC-intensification rate is almost 50% higher over regions with barrier layers, compared to regions without. The presence of a preexisting ocean barrier layer can limit the effects of wind driven vertical mixing and the near-surface cooling response (Wang et al. 2011). This is the case, as while salinity increases from the bottom of the density based mixed layer to the bottom of the barrier layer, temperature does not change much in the barrier layer, resulting in a more favorable ocean state for e.g. the maintenance of a tropical cyclone, as the mixed layer and thermocline are decoupled. A long-term freshening of the upper ocean also tends to intensify the strongest TCs of the western North Pacific, as the increase in stratification reduces their ability to cool the upper ocean (Balaguru et al. 2016).

In this work, we use Argo float observations (Argo, 2000; Roemmich et al., 2003, 2009) to study TC-induced changes in upper ocean properties, focusing on hurricane-strength TCs, i.e., cyclones with maximum sustained winds greater or equal to 64 knots. We describe these changes and their uncertainties using the method by Hu et al., 2024 and compare cases where the pre-event salinity profile increases versus decreases with depth, i.e. we compare two different initial conditions. Our goal is to describe the association between the "increasing" versus "decreasing" vertical structures of the pre-event salinity profile and changes in upper ocean salinity and density during hurricane-strength wind events, which has potential implications for upper ocean stratification and air-sea exchanges during and after the event. If pre-event salinity increases with depth, wind-induced vertical mixing will result in an increase of near-surface salinity, as saltier waters from below are mixed in (Figure 1b); if pre-event salinity decreases with depth, wind-induced vertical mixing will result in a decrease of near-surface salinity, as fresher waters from below are mixed in (Figure 1d). The effect will be larger for pre-event salinity profiles with larger vertical gradients. The two different types of pre-event salinity profiles (Figure 1b, d; Figure S1a in the Supporting Information S1) and associated near surface salinity changes cannot be captured when analyzing the presence versus absence of a pre-event barrier layer and the composite pre-event profiles with and without a barrier layer both show a vertical structure that increases with depth (Figure S1b in the Supporting Information S1). Differences in near

surface salinity changes for the "increasing" and "decreasing" case result in opposite contributions to the density changes with the weather event, with potential implications for air-sea interactions during and after the event. As part of this study, we compare Argo-based results for hurricane-strength TCs with the upper ocean response to hurricane-strength wind events in the HYCOM ocean reanalysis (Chassignet et al., 2007). The HYCOM reanalysis has been used in the past to investigate upper ocean physical and biological processes during hurricane-strength wind events (e.g., Gierach et al., 2009; Prasad and Hogan, 2007; Zamudio and Hogan, 2008) and complements our analysis as it provides time series for each event of interest, instead of sparse pairs of oceanic profiles before and after the weather event, like in the case of Argo observations. We find that results from both Argo and HYCOM are consistent with the vertical mixing of salinity playing a role in how upper ocean stratification changes with the TC passage. This is the case also for hurricane-strength wind events in general, as shown using HYCOM to investigate composites from only hurricane-strength wind events that are not co-located with observed tropical cyclones. Finally, we show that our results for hurricane-strength wind events do not change when we consider only pre-event "increasing" profiles with or without a barrier layer.

→ Mentions of the new figure in the "Methods" section:

(Line 166 in the version of the pdf with tracked changes) We group selected events based on the pre-event vertical structure of salinity from the HYCOM reanalysis, i.e., the vertical structure 2 days before the event: "increasing" events are located where pre-event absolute salinity increases between the density based mixed layer and 50m below the mixed layer (e.g. Figure 1b); "decreasing" events, where salinity decreases between the mixed layer and 50m below (e.g. Figure 1d). The number of "increasing" events is much larger than the number of "decreasing events", with the latter mostly located in the Northern Hemisphere (Figure 2). We use 50 meters for the thickness of the layer considered below the mixed-layer, as it captures the 90th percentile (across events) of the observed mixed layer deepening (not shown). We find our conclusions in the following do not change with the thickness of the layer considered below the mixed-layer, e.g., if we consider 20m to 70m below the mixed-layer.

(Line 210 in the version of the pdf with tracked changes) As for HYCOM (and differently from Hu et al., 2024) we group Argo profile pairs based on the vertical structure of the Argo salinity profile before the weather event: "increasing" pairs include Argo pre-TC profiles with upper ocean salinity increasing with depth (e.g. Figure 1b); "decreasing" pairs include Argo pre-TC profiles with salinity decreasing with depth (e.g. Figure 1d). For each of the two groups, we estimate upper ocean changes with the TC passage and compare them to one another to characterize differences associated with the pre-event vertical structure of salinity. As for HYCOM, there are many more instances for the "increasing" case compared to the "decreasing case" (Figure 2-c, d). We note that, while the total counts in Figure 2 (white box on the top left of each map) are comparable between HYCOM and Argo, the numbers have different meanings. For HYCOM, the number indicates the count of all selected weather events, hence the number of continuous time series of ocean temperature/salinity available for the analysis, as the

model output is available at the location of the weather event at all times of the model simulation. For Argo, the number indicates the count of all available profile pairs to estimate the upper ocean response to the TC passage, as we only have sparse observations from Argo floats.

**Figure 1:** (a, c) Paths of two tropical cyclones and location of Argo profiles close in space and time to the TC tracks and collected before (red marker) and after (purple marker) the TC passage. (b, d) Comparison between salinity profiles collected before (red line) versus after (purple line) the TC passage. One of the pre-event salinity profiles increases with depth (panel b), the other decreases with depth (panel d).

[Figure]

**Figure S1:** Composite vertical structure of pre-event salinity profiles from the HYCOM ocean reanalysis (a) with salinity increasing (red line) versus decreasing (blue) with depth, and (b) with BL (orange) versus without BL (green). While the vertical average has been removed from each profile before calculating composite vertical structures, the details of how salinity increases or decreases with depth for individual profiles in each group are different as these profiles are from different regions of the ocean and different months of the year. Hence, examples in Figure 1b, c may be more helpful than panel (a) here, to visualize differences between the "increasing" versus "decreasing" case.

[Figure]

**3) A second point of confusion is related to the inclusion of non-TC weather events with TC wind speeds. These events are included in the analysis, but there is no discussion of their characteristics, prevalence, or distribution relative to TCs. A brief overview of non-TC high wind events should be given in the Introduction, and their distribution should be indicated somehow in Fig 1, or perhaps in a table that lists percentages of TC and non-TC wind events for each region shown in Fig. 1. Finally, their effects on ocean T, S, and potential temperature should be compared and contrasted to TCs, as they aren't discussed in the latter parts of the paper.**

Thanks for your comments and suggestions. We added text in the "Methods" and "Results and discussion" sections to discuss the comparison between all hurricane-strength wind events and only events that are not co-located with observed tropical cyclones, i. e. that are not within a 200 km radius and +-1 day of an observed TC. New text can be found in the following, after the next paragraph and related figures.

In our response to this reviewer's comment, we include here a direct comparison between the results in the original submission (first figure below) and a version of the plots only considering events that are not co-located with observed tropical cyclones (32% of the total; second figure below, which is new and will be included in the Supporting Information S1 as Figure S4). Results for upper ocean salinity and density in the new version of the panels are consistent with the original results (see top and bottom rows in each figure). For temperature, upper ocean differences between the "increasing" and "decreasing" case are not significant in the new figure (panel f in the second figure below). We note that factors such as wind strength and duration of the strongest winds are key to understand the upper ocean temperature response to weather events, i.e. storm intensity, size and translation speed affect the characteristics of the upper ocean temperature response to the event (Anthes and Chang, 1978; Emanuel and Nolan, 2004; Lin et al., 2017; Samson et al., 2014; Wang et al., 2016; Zhu and Zhang, 2006). Hence, isolating the difference in cooling between the increasing and decreasing case from other relevant factors may be difficult with a limited number of events and it is not a focus here. The discussion of temperature is only included as relevant to better understand results for density.

→ Plot summarizing results included in the original submission of the manuscript, i.e. for all the hurricane-strength wind events. This figure combines panels for composite upper ocean changes in salinity (top), temperature (middle), and potential density (bottom) from figures in the original manuscript that was submitted (Figure 2, S1, 3 in the original submission):

[Figure]

→ A version of the figure above, now only considering events that are NOT co-located with observed tropical cyclones (this figure is included in the Supporting Information S1 for the revised manuscript as Figure S4):

**Figure S4:** Upper ocean salinity (a, b, c), temperature (d, e, f) and potential density (g, h, i) changes during hurricane-strength wind events that are NOT co-located with observed tropical cyclones. Composite changes are shown based on the HYCOM reanalysis for regions where pre-event upper ocean salinity increases (a, d, g) vs decreases (b, e, h) with depth. Panels (c, f, i) show the difference between the "increasing" case (i.e., panel a, d, g) and the "decreasing" case (i.e., panel b, e, h). Dots indicate statistically significant values (95% confidence limit).

[Figure]

Please find in the following new text that will be included in the manuscript. Figure numbers refer to the updated set of figures. Please note that a lighter gray font indicates text in the original submission, still included here for context:

→ New text in the "Introduction", about hurricane-strength wind events that are not co-located with observed tropical cyclones:

(to include at line 31 in the original submission) We note that this wind driven vertical mixing is an important mechanism regulating upper ocean changes for all high wind events, including those that are not associated with tropical cyclones (Cardona et al., 2012, Kuang et al., 2011, Large et al., 1994, Meng et al., 2020).

…

(included in the updated text for the introduction, as shown above) We find that results from both Argo and HYCOM are consistent with the vertical mixing of salinity playing a role in how upper ocean stratification changes with the TC passage. This is the case also for hurricane-strength wind events in general, as shown using HYCOM to

investigate composites from only hurricane-strength wind events that are not co-located with observed tropical cyclones.

→ New text in the "Methods" section, to provide more information regarding hurricane-strength wind events that are not co-located with observed tropical cyclones.

We then proceed in selecting subsequent events continuing with the strongest hurricane-strength wind event in the remaining wind data in the region: using this method, selected events are further than 7 days and ±2° from one another. We find that 32% of the 1238 events identified here are not co-located with an observed tropical cyclone, i.e. are not within a 200 km radius and +-1 day of an observed TC (henceforth referred to as non-TC events); most of these non-TC events are found in the North Atlantic and Pacific, with 35% in the North Atlantic box (in red in Figure 1-a,b) and 56% in the northernmost box in the Pacific (in red in Figure 1-a,b).

→ New text in the "Results and discussion", to mention results for hurricane-strength wind events that are not co-located with observed TCs:

Section 4.1:

Also, upper ocean temperature differences between the "increasing" and "decreasing" case are not  significant when only considering non-TC hurricane-strength wind events (Figure S4). We note that factors such as wind strength and duration of the strongest winds are key to understanding the upper ocean temperature response to …

…

The difference between the two composites is statistically significant (Figure 2-c). This is also seen when only locations with no pre-event barrier layer are considered (Figure S3-a, b, c in Supporting Information S1). Additionally, the same result is found when only non-TC hurricane-strength wind events are included in the composites (Figure S4).

…

As for temperature and salinity, similar results are found for the difference between "increasing" versus "decreasing" composites for density when only locations with no pre-event barrier layer are considered (Figure S3-g, h, i in Supporting Information S1). This is the case also when only  non-TC hurricane-strength wind events are included in the composites (Figure S4).

**Analysis methods: according to Section 3.2.4 (lines 175-183), T and S anomalies with respect to TCs are computed by subtracting the mean annual cycle of non-TC-classified profiles from TC profiles. First of all, it is not clear how the mean seasonal cycle is computed: is it computed for each 1x1 grid box or averaged over the boxes shown in Figure 1?**

Thank you for your question. As in Hu et al. (2024), we estimate the seasonal cycle on a 1x1 degree grid using a local linear regression model. We now include this description in the revised manuscript (in the "Methods" section).

**Second, by subtracting the non-TC mean profile, what's left is a profile that is potentially unique to the TC environment. Third, I wonder if it would be more informative to subtract the T or S profile averaged from days -2 to 0 from all days plotted in Figure 2 and similar figures. The reason being that these figures show T and S anomalies whose vertical distribution is nearly constant with time, which might simply reflect the classification scheme and not wind-generated vertical mixing effects. Subtracting the pre-event averaged anomaly might show changes in the DEPTH of the anomalies with TC passage, which is the expected signal.**

Thank you for your comment. In the following figures, we show a version of Figure 2, 3, S2 subtracting (from each timestep) the time average between -2 and 0 days (instead of subtracting the pre-event profile at -2 days as in the original submission of our manuscript and in Hu et al. 2024). These figures are similar to figures in the original submission, hence we only include them here.

→ Plot summarizing results when subtracting the time average between -2 and 0 days (instead of subtracting from each timestep the pre-event profile at -2 days as in the original submission of our manuscript).

[Figure]

**Use of supplemental figures to discuss temperature changes: The authors dedicate a fair amount of text to discussing temperature changes for increasing and decreasing salinity profiles (i.e., lines 195-221, but the discussion refers to two multipanel plots of supplemental figures. I can understand the desire to limit the number of figures, but assessing changes to ocean stratification requires consideration of both salinity and temperature, so it might be better to combine**

**these changes into one plot by adding temperature differences as contours to Figures 2 and 4.**

Thanks for the suggestion. We have included contours for the increasing versus decreasing differences in temperature in the plot for density (updating panel c in Fig. 3 and 5 in the original submission). The new panel for Figure 3 and Figure 5 are included below: black contours are temperature differences and are shown only when significant. A continuous line indicates positive values, a dashed line indicates negative values. We now refer to the new panels (Figure 4-c and 6-c) in the "results and discussion" sections.

HYCOM (Figure 4-c)

[Figure]

Argo (Figure 6-c)

[Figure]

**Figure improvements: most figures could be improved through more informative panel labels. For instance, Figure 1 panel labels should include "HYCOM" in panels a and b, and "dS/dz >0" and "dS/dz<0" to help readers understand what they're looking at ("d" should be a partial derivative symbol). Figures 2-5 should include panel labels "dS/dz>0", "dS/dz<0", and "difference (a-b)."**

Thanks for the suggestions. We improved the figure labels throughout the paper. Examples are shown below for Figure 1-a and Figure 2 in the original submission, i.e., Figure 2-a and 3 in the new version of the manuscript:

[Figure]

[Figure]

**This work has the potential to be quite informative to the TC and air-sea interaction communities and I hope that the authors can incorporate some of these suggestions to produce a paper that should be widely read among these communities.**

Thanks again for all your helpful suggestions!

**Review 2**

The authors would like to thank the reviewer for their thoughtful comments and suggestions. Our responses are below (reviewer's comments are indicated with black bold font).

**In this paper the authors discussed the trend of ocean salinity and density profiles before and after tropical cyclones using Argo float observations and HYCOM ocean reanalysis. The upper ocean changes and the vertical structures are presented, but all figures are discussing the pre-event salinity increases and decreases with depth without much background information. There is even no example showing how much the pre-event salinity changes. That being said, the results are very well presented and the manuscript is very well written. In my opinion, the paper can be published after addressing a few minor issues.**

Thank you for your comments. Each was appreciated and led to new text/figures that are described in the following.

**Here is the list of my concerns:**

**Line 65. Why do the authors look at the vertical structures of the ocean under tropical cyclone conditions under pre-event salinity changes? The authors should mention and discuss this more explicitly in the introduction. Do you want to investigate the response of the ocean under different initial conditions?**

**Line 126. Can the authors show an example of increasing and decreasing cases?**

Thanks for your comments and suggestions. New text and figures were included to clarify the rationale for our analysis.

New text and figures can be found in the following. Figure numbers refer to the updated set of figures. Please note that a lighter gray font indicates text in the original submission (even if edits were made in response to other reviewer's comments) and is included here for context:

→ New text in the "Introduction" to clarify the rationale for our analysis, as well as clarify that we also analyze the effect of the presence of a barrier layer on our results (the new text will appear in the revised manuscript starting at line 60 of the original pdf):

Also, Balaguru et al. (2012) shows that the TC-intensification rate is almost 50% higher over regions with barrier layers, compared to regions without. The presence of a preexisting ocean barrier layer can limit the effects of wind driven vertical mixing and the near-surface cooling response (Wang et al. 2011). This is the case, as while salinity increases from the bottom of the density based mixed layer to the bottom of the barrier layer, temperature does not change much in the barrier layer, resulting in a more favorable ocean state for e.g. the maintenance of a tropical cyclone, as the mixed layer and thermocline are decoupled. A long-term freshening of the upper ocean also tends to

intensify the strongest TCs of the western North Pacific, as the increase in stratification reduces their ability to cool the upper ocean (Balaguru et al. 2016).

In this work, we use Argo float observations (Argo, 2000; Roemmich et al., 2003, 2009) to study TC-induced changes in upper ocean properties, focusing on hurricane-strength TCs, i.e., cyclones with maximum sustained winds greater or equal to 64 knots. We describe these changes and their uncertainties using the method by Hu et al., 2024 and compare cases where the pre-event salinity profile increases versus decreases with depth, i.e. we compare two different initial conditions. Our goal is to describe the association between the "increasing" versus "decreasing" vertical structures of the pre-event salinity profile and changes in upper ocean salinity and density during hurricane-strength wind events, which has potential implications for upper ocean stratification and air-sea exchanges during and after the event. If pre-event salinity increases with depth, wind-induced vertical mixing will result in an increase of near-surface salinity, as saltier waters from below are mixed in (Figure 1b); if pre-event salinity decreases with depth, wind-induced vertical mixing will result in a decrease of near-surface salinity, as fresher waters from below are mixed in (Figure 1d). The effect will be larger for pre-event salinity profiles with larger vertical gradients. The two different types of pre-event salinity profiles (Figure 1b, d; Figure S1a in the Supporting Information S1) and associated near surface salinity changes cannot be captured when analyzing the presence versus absence of a pre-event barrier layer and the composite pre-event profiles with and without a barrier layer both show a vertical structure that increases with depth (Figure S1b in the Supporting Information S1). Differences in near surface salinity changes for the "increasing" and "decreasing" case result in opposite contributions to the density changes with the weather event, with potential implications for air-sea interactions during and after the event. As part of this study, we compare Argo-based results for hurricane-strength TCs with the upper ocean response to hurricane-strength wind events in the HYCOM ocean reanalysis (Chassignet et al., 2007). The HYCOM reanalysis has been used in the past to investigate upper ocean physical and biological processes during hurricane-strength wind events (e.g., Gierach et al., 2009; Prasad and Hogan, 2007; Zamudio and Hogan, 2008) and complements our analysis as it provides time series for each event of interest, instead of sparse pairs of oceanic profiles before and after the weather event, like in the case of Argo observations. We find that results from both Argo and HYCOM are consistent with the vertical mixing of salinity playing a role in how upper ocean stratification changes with the TC passage. This is the case also for hurricane-strength wind events in general, as shown using HYCOM to investigate composites from only hurricane-strength wind events that are not co-located with observed tropical cyclones. Finally, we show that our results for hurricane-strength wind events do not change when we consider only pre-event "increasing" profiles with or without a barrier layer.

→ Mentions of the new figure in the "Methods" section:

(Line 166 in the version of the pdf with tracked changes) We group selected events based on the pre-event vertical structure of salinity from the HYCOM reanalysis, i.e., the vertical structure 2 days before the event: "increasing" events are located where

pre-event absolute salinity increases between the density based mixed layer and 50m below the mixed layer (e.g. Figure 1b); "decreasing" events, where salinity decreases between the mixed layer and 50m below (e.g. Figure 1d). The number of "increasing" events is much larger than the number of "decreasing events", with the latter mostly located in the Northern Hemisphere (Figure 2). We use 50 meters for the thickness of the layer considered below the mixed-layer, as it captures the 90th percentile (across events) of the observed mixed layer deepening (not shown). We find our conclusions in the following do not change with the thickness of the layer considered below the mixed-layer, e.g., if we consider 20m to 70m below the mixed-layer.

(Line 211 in the version of the pdf with tracked changes) As for HYCOM (and differently from Hu et al., 2024) we group Argo profile pairs based on the vertical structure of the Argo salinity profile before the weather event: "increasing" pairs include Argo pre-TC profiles with upper ocean salinity increasing with depth (e.g. Figure 1b); "decreasing" pairs include Argo pre-TC profiles with salinity decreasing with depth (e.g. Figure 1d). For each of the two groups, we estimate upper ocean changes with the TC passage and compare them to one another to characterize differences associated with the pre-event vertical structure of salinity. As for HYCOM, there are many more instances for the "increasing" case compared to the "decreasing case" (Figure 2-c, d). We note that, while the total counts in Figure 2 (white box on the top left of each map) are comparable between HYCOM and Argo, the numbers have different meanings. For HYCOM, the number indicates the count of all selected weather events, hence the number of continuous time series of ocean temperature/salinity available for the analysis, as the model output is available at the location of the weather event at all times of the model simulation. For Argo, the number indicates the count of all available profile pairs to estimate the upper ocean response to the TC passage, as we only have sparse observations from Argo floats.

**Figure 1:** (a, c) Paths of two tropical cyclones and location of Argo profiles close in space and time to the TC tracks and collected before (red marker) and after (purple marker) the TC passage. (b, d) Comparison between salinity profiles collected before (red line) versus after (purple line) the TC passage. One of the pre-event salinity profiles increases with depth (panel b), the other decreases with depth (panel d).

[Figure]

**Figure S1:** Composite vertical structure of pre-event salinity profiles from the HYCOM ocean reanalysis (a) with salinity increasing (red line) versus decreasing (blue) with depth, and (b) with BL (orange) versus without BL (green). While the vertical average has been removed from each profile before calculating composite vertical structures, the details of how salinity increases or decreases with depth for individual profiles in each group are different as these profiles are from different regions of the ocean and different months of the year. Hence, examples in Figure 1b, c may be more helpful than panel (a) here, to visualize differences between the "increasing" versus "decreasing" case.

[Figure]

**Why use 50 meters below the mixing layer?**

Thanks for your question. We use 50 meters for the thickness of the layer considered below the mixed-layer, as it captures the 90th percentile (across events) of the observed mixed layer deepening (see plot below showing the time evolution of the 90th percentile mixed-layer increase across events). We also confirmed that our results do not change if we use a thickness between 20 and 70m.

We made edits to the text in the "Methods" section to clarify this point.

→ This figure shows the time evolution of the 90th percentile mixed layer increase across events.

[Figure]

**90th percentile mixed layer increase across events**

[Figure]

Hours since the event

**How is the mixing layer determined in this work? What will happen if the differences for salinity or density are very small?**

Thanks for your questions. Mixed layer properties are estimated using the methodology described in Holte and Talley (2009), i.e. a hybrid algorithm that models the general shape of each profile, searches for physical features in the profile, and calculates threshold and gradient MLDs to assemble a suite of possible MLD values, before selecting a final MLD estimate. This method has been shown to work well also in regions where the mixed layer exhibits great variability and the estimate of the MLD is overall challenging, e.g. winter mixed layers north of the Subantarctic Front, which can reach depths of 500 m and blend into deeper waters and remnant mixed layers (Holte et al. 2009). [This text is now included at the end of Section 2.3]

If differences in salinity between the ML and layers below are very small, waters mixed vertically have similar salinity, and the effect of wind induced vertical mixing on upper ocean salinity changes may be masked by other processes. Uncertainties in our analysis show that we observe a significant signal that is consistent with the effects of wind driven vertical mixing. A note is now included in the "Introduction" to clarify that the effect of the wind induced vertical mixing of interest is larger for events with pre-event salinity profiles characterized by larger vertical gradients.

**Line 145. I think this coordinate system is only used for Fig. 6 and the definition of cross-track angle is confusing. From Hu et al. 2024, isn't it the shortest distance between Argo float and cyclone track? The word "angle" in Fig. 6 is also confusing because it may refer to some angles between -180/pi to 180/pi. Is there any reason for not using the distance in km or meters?**

Thank you for your comment. The cross-track angle is used as it offers a more accurate spherical model for measuring the distance from the TC track, compared to e.g. using distance in km. The cross-track angle is calculated by determining the angle between

line segments from the sphere's center to the two points on its surface and it is equivalent to the haversine distance. When longitude is constant, the induced angle simplifies to the absolute difference in latitude. Conversely, when latitude is constant, the angle simplifies to the absolute difference in longitude, considering the cosine of latitude.

This information is now included in the "Methods" section, i.e.

The x-coordinate is the cross-track angle, which is determined by calculating the angle between line segments from the sphere's center to two surface points, equivalent to the haversine distance. When longitude is constant, this angle is the absolute difference in latitude. When latitude is constant, it is the absolute difference in longitude, adjusted by the cosine of the latitude. The cross-track angle aligns with the great circle distance and is used here as it provides a more accurate spherical model for measuring the distance from the TC track, compared to e.g. using distance in km.

**Line 150. How the Argo data and HYCOM data being used is still confusing to me. It seems that most figures are plotted directly using HYCOM data, but the pre-event conditions are determined based on the Argo floats?**

Thanks for your comment. The pre-event conditions in the Argo analysis are based on Argo profiles; the pre-event conditions in the HYCOM analysis are based on HYCOM profiles. We included clarifications in Section 3.1 and 3.2.2 in the manuscript.

**Line 190. More technical details are needed for the statistical analysis. Are pre-TC and post-TC profiles having the same standard deviations?**

As in Hu et al. 2024, only the non-tc activity is represented here as a random process, hence, the variance of the before and after profiles is the same under this model. We have added this clarification to Section 3.2.5.

**Figure 2a. Any interpretation of the alternative blue red blue colors between 50 to 200 m from day -1 to day 2?**

By construction, the peak wind for each event is at day zero, hence we expect to see the effects of a deeper and stronger mixing just after that. In the "increasing" case, this may result in the salinity increase reaching deeper (depending on the vertical structure of the salinity profile). Depending on how the wind increases towards the peak and what the details of the pre-event profile are, we may see the effect of mixing also before the peak wind. In the "increasing" case, this may result in a decrease in salinity at depth, as less saline waters are mixed downward. Finally, as the wind reduces after the peak, the mixing weakens again. As the effects of vertical mixing depend on both the strength of the wind and the vertical structure of the salinity profile, whether we see the sign of the composite signal alternating in time will depend on the timing and location of the events in the composite, as the vertical structure of the salinity profile changes by region and time.

**How about the decrease of the density at day 0 in Figure 3a?**

The decrease in subsurface density at day 0 is consistent with observed changes in temperature and salinity. As discussed in the previous answer, the peak wind for each event is at day zero, hence we expect to see the effects of this mixing just after that. Yet, depending on how the wind increases towards the peak and what the details of the pre-event profile are, we may see the effects of mixing also just before/at day 0.

**If Figure 4 and Figure 2 are presenting the same result, why are the Argo data having much stronger increase at day 0?**

While the overall difference in upper ocean salinity changes between the "increasing" and "decreasing" case is consistent between HYCOM and Argo, the two products show different amplitudes for the signal of interest (Figure 3c versus 5c in the version of the pdf with tracked changes), which may be related to e.g. how hurricane-strength wind events and related vertical mixing processes in the ocean are represented in the HYCOM model, and what the availability is of sparse ocean observations co-located with events of interest.

A clarification was included at line 332 in the version of the pdf with tracked changes:

This difference may be related to how vertical mixing processes are represented in the HYCOM model, the availability of sparse ocean observations co-located with events of interest, as well as the details of the vertical structure of pre-event upper ocean properties in the model versus observations.

Thanks again for all your helpful suggestions!

---

## Author Response (AR2)

**Review 1**

The authors would like to thank the reviewer for their helpful suggestions. Our responses are below (reviewer's comments are indicated with black bold font).

**L21: change "more and more powerful" to "more powerful"**

Done.

**L28. Change colon to period and begin a new sentence.**

Done.

**L29: change "tends to a decreasing" to "decreases"**

Done.

**L32: Delete "We note that"**

Done. We also deleted "this" and now the sentence reads "Wind driven vertical mixing is an important mechanism regulating upper ocean changes for all high wind events, including those that are not associated with tropical cyclones (Cardona and Bracco, 2012; Kuang et al., 2011; Large et al., 1994; Meng et al., 2020)."

**L35: change "upper ocean changes in temperature" to "changes in upper ocean temperature" to mirror "changes in upper ocean salinity" later in the same sentence.**

Done.

**L40: replace ", regulating the strength of" with "and"**

Done.

**L41-42: by "taking into account" do you mean "assimilation of"?**

Balaguru et al 2020 show a strong inverse relationship between salinity and tropical cyclone (TC) rapid intensification (RI) in the eastern Caribbean and western tropical Atlantic due to near-surface freshening from the Amazon–Orinoco River system, i.e. salinity stratification reduces SST cooling, which contributes to stronger surface enthalpy flux for rapidly intensifying TCs. Balaguru et al 2020 statistical results are confirmed through experiments with an ocean mixed layer model, which show that the salinity-induced reduction in SST cold wakes increases significantly as the storm's intensification rate increases (the impact of salinity on more weakly intensifying storms is insignificant). Also, through experiments with a statistical RI prediction scheme, it is

found that the inclusion of surface salinity significantly improves the RI detection skill, offering promise for improved operational RI prediction. This is why we used the wording "taking into account".

**L42-43: delete "also it may be important to better understand post-cyclone air-sea interactions."**

Done.

**L47: replace "…(Sun et al. 2021). Contrarily, in the Southern Hemisphere a salty wake after TC passage is observe on" with "…(Sun et al. 2021) and the left-hand side of the storms in the Southern Hemisphere."**

Done.

**L50: this is improper use of a semi-colon. From https://www.niu.edu/writingtutorial/index.shtml: "Use a semicolon to join two related independent clauses in place of a comma and a coordinating conjunction (and, but, or, nor, for, so, yet). Make sure when you use the semicolon that the connection between the two independent clauses is clear without the coordinating conjunction." The two phrases joined by a semicolon here should be two separate sentences. Authors should check other uses of semicolon throughout text for other possible edits.**

Done and checked the use of semicolons all over the manuscript.

**L51: delete "i.e, in most regions where cyclonic activity is found"**

Done.

**L59: delete "e.g."**

Done.

**L77-78: the phrase "which has potential implications for upper ocean stratification and air-sea exchanges" and its close relatives seems to be overused. It only needs to be highlighted once in the introduction and once in the conclusions.**

Done.

**Review 2**

The authors would like to thank the editor for their helpful suggestions. Our responses are below (editor's comments are indicated with black bold font).

**Several data sets and data products are used and the authors show mention these correctly in the data availability section. I would like to ask the authors to check whether the data originators request a separate data use statement, or possibly ask for citing of specific papers describing the data.**

Requested data statements are now included in the acknowledgements.

**The barrier layer plays an important role in the text. However, there is no definition of it in the text. It would be useful to define it and its characteristics, so that all readers think about the same feature when barrier layer is mentioned.**

This sentence is now included in the introduction at line 58 in the pdf highlighting differences from the previous version:

Using Argo observations and a composite approach, Steffen and Bourassa (2018) quantify barrier layer development due to tropical cyclones. A barrier layer is the salinity-stratified isothermal layer situated between the base of the mixed layer and the top of the thermocline (Godfrey et al., 1989), in some regions of the ocean, and it acts as a barrier to the turbulent entrainment of cold thermocline water into the surface mixed layer (Cronin et al., 2002).

**L9 Add "surface" before temperature for clarity**

Done.

**L42 delete (Balaguru et al., 2021) as it is already mentioned earlier in the sentence**

Done.

**L43 "it is important" instead of: it may be important**

This sentence was removed, as suggested by reviewer #1.

**L61 … by about 35%**

Done.

**L74 Please add SI units for knots; they could be inserted in parentheses**

Done.

**L98-99 "Data and methods used in our analysis are described in Section 2 and 3, respectively. Results are presented in Section 4. Section 5 includes a summary**

and conclusions." This can be deleted as the structure of the manuscript shows up in the text below and these descriptions do not add any new info.

Done.

L107-108 Change to (note the punctuation): "Data are reported at six-hourly resolution, since 1851 for the Atlantic hurricane database (HURDAT2), since 1949 for the Northeast and North Central Pacific (HURDAT2) and since 1945 for the JTWC Best Track Data."

Done.

L139 Please add SI units for knots; they could be inserted in parentheses

Done.

L142-143 "… we exclude, from the remaining data, any other event within ±2∘ and 7 days before and after the one selected." Maybe add a sentence why you do this.

This sentence is now included at line 148 in the pdf highlighting differences from the previous version:

… we exclude, from the remaining data, any other event within ±2∘ and 7 days before and after the one selected, to ensure selected events are independent.

L164 instead of: "e.g. Figure 3, 4", see Figures 3 and 4

Done.

L193-195 This can be more succinct: … which we also call the "signal" profile, needs to be between 2 days before and 20 days after the TC passage to hedge against storm effects prior to the event, as described in Hu et al. (2024) and consistent with Cheng et al. (2015).

Done.

L228 change to: … as shown in Figures 5 and 6.

Done.

L233 I think "of upper ocean changes during hurricane-strength TCs" can be deleted, as this is mentioned again in the next line

Done.

L246-248 "A difference in cooling is also seen when comparing HYCOM composites based on the vertical structure of salinity before a hurricane-strength wind event as in our analysis, i.e., pre-event salinity "increasing" versus

**"decreasing" with depth, instead of pre-event barrier layer present versus not."**
**This sentence is hard to understand. Please rephrase for better reading**

Done. Now the sentence reads:

A difference in cooling is also observed when comparing HYCOM composites based on the vertical structure of salinity before the event, i. e., pre-event salinity "increasing" versus "decreasing" (instead of pre-event barrier layer present versus not).

**L265 At line 133 it is written that absolute salinity is used (for Argo). Here 0.05 psu is given? Which one do you use? Please note that psu is not a unit.**

Our results for salinity are reported as psu (we include "psu" to make it clear that what is being reported is based on the practical salinity scale). We use Argo absolute salinity to calculate potential density.

**L300-301 "The analysis of HYCOM fields in the previous section quantifies upper ocean changes with hurricane-strength wind events based on the pre-event vertical structure of salinity." This sentence can be deleted as it is clear from the previous section.**

Done.

**L303-305 "In this section, we focus on the specific case of hurricane-strength TCs and analyze Argo profile data to estimate upper ocean changes based on pre-event salinity "increasing" versus "decreasing" with depth." This can be deleted as the title of the section gives this info and the following text is showing it.**

Done.

**L311-314 "This difference may be related to how vertical mixing processes are represented in the HYCOM model, the availability of sparse ocean observations co-located with events of interest, as well as the details of the vertical structure of pre-event upper ocean properties in the model versus observations." This is quite general information, which then do not convey much understanding to the reader. Please explain and elaborate how such factors might affect the difference.**

This sentence is now included at line 319 in the pdf highlighting differences from the previous version:

This difference may be related to e.g. how vertical mixing processes are represented in the HYCOM model, and in particular how deep the wind-induced mixing reaches with the event; the details of the vertical structure of pre-event upper ocean properties in the model versus observations, i.e. the initial condition of events of interest; the availability of sparse ocean observations co-located with events of interest compared to the continuous time series the HYCOM model provides in its spatial domain.

**L316-320 This one long sentence is hard to understand. Please rephrase and split it into two sentences for better understanding**

These sentences are now included at line 327 in the pdf highlighting differences from the previous version:

However, it is not surprising that the same vertical pattern of changes, with different signs at different depths, is not seen in Figure S5-a in Supporting Information S1. While the near surface cooling is common across locations used to estimate ocean changes for the 'increasing' versus 'decreasing' composite, the depth of the warming signal and its vertical extent vary , (analogous to what discussed for Figure 3-a).

**L328-329 "Consistent with wind-induced vertical mixing, salinity increases versus decreases with the TC passage for the "increasing" versus "decreasing" case (Figure 5)" This sentence is not clear. Please rephrase**

This sentence is now included at line 339 in the pdf highlighting differences from the previous version:

Consistent with wind-induced vertical mixing, salinity increases with the TC passage for the "increasing" case, and it decreases for the "decreasing" case (Figure 5).

**Caption Figure 7: Please mention and define the abbreviations MLT, MLS and MLPD as used as y-axis label in the caption**

Done.

**L447 delete pp.**

Done.

**L543 Add after 30,: 1933, doi:10.1029/2003GL017878**

Done.

**L552 delete pp.**

Done.

**L554 Add after 30,: 1572, doi:10.1029/2002GL016717**

Done.

**L571 delete p. before 1287**

Done.